# Phase Contrast Image-Based Rapid Antimicrobial Susceptibility Testing of Bacteria in Liquid Culture Media

**DOI:** 10.3390/s23010059

**Published:** 2022-12-21

**Authors:** Xiaonan Zhang, Xuenian Wang, Yaqing Bao, Zhiyuan Shen, Yang Xu, Bei Wang, Haowei Zhang, Tian Guan, Yonghong He

**Affiliations:** 1School of Medicine, Tsinghua University, Beijing 100084, China; 2Institute of Optical Imaging and Sensing, Shenzhen Key Laboratory for Minimal Invasive Medical Technologies, Shenzhen International Graduate School, Tsinghua University, Shenzhen 518055, China; 3GBA Center for Medical Device Evaluation and Inspection, National Medical Products Administration, Shenzhen 518045, China; 4Medical Optical Technology R&D Center, Research Institute of Tsinghua, Pearl River Delta, Guangzhou 510700, China; 5School of Life Sciences, Tsinghua University, Beijing 100084, China; 6Department of Physics, Tsinghua University, Beijing 100084, China

**Keywords:** bacteria, rapid AST, phase contrast microscope, MIC

## Abstract

Currently, the world is facing the problem of bacterial resistance, which threatens public health, and bacterial antimicrobial susceptibility testing (AST) plays an important role in biomedicine, dietary safety and aquaculture. Traditional AST methods take a long time, usually 16–24 h, and cannot meet the demand for rapid diagnosis in the clinic, so rapid AST methods are needed to shorten the detection time. In this study, by using an in-house built centrifuge to centrifuge bacteria in a liquid medium onto the inner wall of the bottom surface of a counting plate, and using a phase contrast microscope to track bacterial growth under the effect of different antibiotic concentrations, the results of the minimum inhibitory concentration (MIC) of bacteria under the effect of antibiotics can be obtained in as early as 4 h. We used a combination of *E. coli* and tigecycline and obtained MIC results that were consistent with those obtained using the gold standard broth micro-dilution method, demonstrating the validity of our method; due to the time advantage, the complete set can be used in the future for point of care and clinical applications, helping physicians to quickly obtain the MIC used to inhibit bacterial growth.

## 1. Introduction

Nowadays, the misuse of antibiotics has led to a gradual increase in bacterial resistance, especially in developing countries [1], increasing the incidence of globalized infectious diseases and potentially leading to a global situation of no drug availability [2,3], threatening public health and increasing healthcare costs [4]. Therefore, countries around the world have begun to focus on the restriction of antibiotic drugs, and antimicrobial susceptibility testing (AST) of bacteria is an effective way to prevent antibiotic abuse. Reliable and rapid AST methods become especially important for giving timely and effective therapeutic measures [5,6], otherwise inappropriate use of antibiotics by the concerned personnel could potentially breed superbugs [7].

Traditional methods of AST mainly include the disk diffusion method, the broth dilution method, etc. However, the time required for these methods is generally 16–24 h [8]. Phase contrast imaging converts phase differences to amplitude differences by interferometry [9], eliminating the need for sample labeling, reducing damage to the sample and offering the possibility of in situ imaging of biological cells and tissues in vivo [10]. A number of rapid AST methods based on optical imaging have emerged in recent years, and image-based methods not only allow for the minimum inhibitory concentration (MIC), but also for the morphology of bacteria [11,12,13]. The current image-based rapid AST methods are mainly divided into two categories: one is to use agarose to make solid media to immobilize bacteria [14,15,16,17,18] and then use a microscope to track their growth; however, this method is less adequate than liquid media in terms of both nutrition and antibiotic exchange. The other is liquid media combined with microfluidic devices to allow bacteria to incubate continuously in droplets [19] or in channels [20,21], and then use a microscope to track their growth; however, these microfluidic devices require micro-nano fabrication, are complicated to make and are costly, which is not conducive to their application in clinical settings.

Both solid media and microfluidic devices are designed to limit the movement of bacteria and keep them in the focal plane of the imaging system at all times, thus tracking the continuous growth of bacteria in real time. Where the solid medium holds the bacteria in place, the microfluidic approach confines the bacteria in a small space. In this study, we used centrifugation to downscale the three-dimensional bacterial distribution to two dimensions [22], so that the bacteria in the bacteria solution settled on the bottom inner wall of the counting plate, on the focal plane of the phase contrast imaging system. We found that the vast majority of bacteria settled at the bottom of the counting plate, which reduces the impact of floating bacteria settling when tracking bacterial growth. We used an in-house built centrifuge to centrifuge bacteria to the bottom of the counting plate, and then used an in-house built phase contrast microscope to track and photograph the bacteria under the effect of different antibiotic concentrations at 1 h intervals. The MIC was derived by counting the area of bacteria and calculating the bacterial growth rate, and then based on the trend of bacterial growth rate. To verify the validity of our method, we used *E. coli* and tigecycline for the experiments, and used the gold standard broth micro-dilution method as a control experiment, and the final MIC results obtained by our method were consistent with those of the broth micro-dilution method. 

In this study, we used a centrifuge to settle the bacteria in the bacteria solution to the inner wall of the counting plate so that the bacteria were in the focal plane of the phase contrast microscope, and obtained the bacterial growth rate curve and finally the AST results through image processing. We compared the traditional AST methods, existing rapid AST methods and our method (Appendix A). Our method enables the rapid AST of bacteria in liquid culture media, which can meet the needs for a rapid acquisition of AST results in clinical diagnostics and point of care testing, while the low cost and low complexity of the centrifuge and phase contrast microscope provide possibilities for practical applications.

## 2. Materials and Methods

### 2.1. Bacterial Incubation and Sample Preparation

Firstly, the recovery solution was added to the lyophilized *E. coli* (25922, Guangdong Huankai Microbial Sci. & Tech. Co., Ltd., Guangzhou, China) using a syringe and mixed well to achieve strain recovery. Then, the liquid medium was prepared according to the ratio of 2.5 g of Luria-Bertani (LB) liquid medium dry powder (L1010, Solarbio Sci.& Tech. Co., Ltd., Beijing, China) and 100 mL of pure water, fully dissolved and sterilized under high temperature and pressure. The recovered bacterial suspension was added to the liquid medium and incubated overnight at 35 °C in an air bath thermostatic oscillator (SHZ-82, Champion Instrument Manufacturing Co., Ltd., Changzhou, China) with uniform shaking at 190–220 rpm.

The drug sensitive inoculation medium (CAMHB turbidity type, G10251, Kangtai Biotechnology Co., Ltd., Wenzhou, China) and tigecycline drug-sensitive reagent plate (J1A079, Kangtai Biotechnology Co., Ltd., Wenzhou, China) were placed in the air bath thermostatic oscillator to restore the temperature. The bacteria solution was removed from the liquid medium and added to sterile water, then turned up to 0.5MCF using a bacterial turbidity meter (WGZ-XT, Yuanhengtong Technology Co., Ltd., Shenzhen, China). The above-mixed bacteria solution was aspirated and mixed thoroughly at a ratio of 1:200 of bacteria solution and drug sensitive inoculation medium, and then 100 μL was added to each well of the selected concentration of tigecycline drug-sensitive reagent plate, in which the antibiotic concentration was selected as 0.064 μg/mL, 0.125 μg/mL, 0.25 μg/mL and 0.5 μg/mL; then, one antibiotic-free well was selected for quality control. Dissolve the antibiotics in the bacterial solution, and then pipette the bacterial solution in the wells to the counting plate for use.

### 2.2. Centrifugal Device and Centrifugation

The bacteria in the bacteria solution are centrifuged to the bottom of the counting plate to achieve enrichment of bacteria on the bottom surface and in the focal plane of the phase contrast microscope, where the centrifugation effect is schematically shown in Figure 1. The centrifuge consists of a motor (DC-BL2845, XuanDong Technology Co., Ltd., Shenzhen, China), a driver (EC30, XuanDong Technology Co., Ltd., Shenzhen, China), a motion controller (iMC404A, Yanwei Technology Co., Ltd., Guangzhou, China) and a rotor, and the rotation is controlled by the motion control software. A counting plate is placed at the edge of the rotor. The rotation of the motor drives the rotor to rotate at high speed and the centrifugal force centrifuges the bacteria in the bacteria solution to the bottom of the counting plate. The counting plate with bacteria solution is placed in the rotor hole of the centrifuge device, and the motor is controlled to rotate at 6400 rpm for 20 s.

The bacteria in the counting plate float in the bacteria solution, and the bacteria will slowly settle by their own gravity over time. After centrifugation, the bacteria in the bacteria solution are rapidly enriched to the inner wall of the bottom surface of the counting plate. Therefore, after centrifugation, the growth of bacteria is continuously photographed with a phase contrast microscope, which can effectively reduce the impact of the increase in the number of bacteria caused by the settling of floating bacteria.

### 2.3. Phase Contrast Microscope

The phase contrast microscope mainly consists of an optical system (Figure 2a) and a motorized translation stage, using an LED (Gree XP-E2 central wavelength 525 nm, 5 W power) as the light source, putting an objective lens (25×,NA 0.4,GCO-2104, Daheng Optics, Beijing, China) to focus the beam and connect a homemade 0.3 mm aperture to simulate a point source, and then using an achromatic lens (f = 30 mm, GCL-010650, Daheng Optics, Beijing, China) to collimate the beam. The light source is paired with a control module that regulates the LED current and is equipped with a heat sink and fan for heat dissipation.

A phase plate was placed under the collimated beam to improve the imaging contrast in conjunction with the phase contrast objective, and then a biconvex lens (f = 25.4 mm, N-BK7, Lubang Photonics Technology Co., Ltd., Changsha, China) was attached to focus the beam. The phase contrast objective (25×, NA = 0.4, Weiteshijie Technology Co., Ltd., Shenzhen, China) and mono camera (ME2L-161-61U3M-L, Daheng Optics, Beijing, China) were placed under the sample stage.

The motorized translation stage consists of linear modules (X:FSK30E200T02, Y:FSL30E75T02/J, Z:FSK30E50T04, Fuyu Technology Co., Ltd., Chengdu, China) to realize the linear round-trip motion, with the driver (FMDD36D22NOM, Fuyu Technology Co.,Ltd., Chengdu, China) and the motion controller (iMC404A, Yanwei Co., Ltd., Guangzhou, China), and to send commands through the motion control software to achieve the precise translation of the sample. The counting plate mount (Figure 2(b2)) allows five counting plates to be placed simultaneously.

Using this phase contrast microscope, we can obtain images of bacteria in the bacteria solution after centrifugation. With the movement of the motorized translation stage, we can obtain images of bacteria in the bacteria solution of five different concentrations of antibiotics, and image the same position of the bacteria solution of five concentrations of antibiotics every one-hour interval, and finally obtain the growth of bacteria with time under the effect of different concentrations of antibiotics.

### 2.4. Image Processing

In order to count the growth of bacteria and combine the characteristics of phase contrast images, we use adaptive thresholding to segment the images and distinguish the bacteria from the background. Since the area of the light source irradiating on the camera’s light-sensitive chip is slightly smaller than the chip size, the background of the obtained image is somewhat uneven in terms of light and darkness. Therefore, before segmentation, the background of the image needs to be subtracted to reduce the effect of the uneven background; then the phase contrast image is Gaussian filtered to eliminate the effect of Gaussian noise; finally, the image is segmented using adaptive thresholding.

Normalized growth rate of bacteria is defined as [15]:(1)$$\mathrm{Normalized}\;\mathrm{growth}\;\mathrm{rate}\;=\frac{A_{t}}{A_{0}}$$
where *A_t_* and *A*_0_ refer to the bacterial area at time t and the initial time, respectively.

The normalization process can eliminate the effect caused by the different number of bacteria in different fields of view at the starting time point, and by calculating the growth rate of bacteria, the growth of bacteria under different concentrations of antibiotics can be obtained, so that the MIC can be obtained according to the bacterial growth curve.

### 2.5. AST Using the Broth Microdilution Method

The broth microdilution method, which is the gold standard in AST [12], was used as a control experiment in this study. We used a tigecycline drug-sensitive reagent plate for the experiment, and judged whether the bacterial growth was inhibited by the turbidity degree of the bacteria solution. Five concentrations of 0, 0.064 μg/mL, 0.125 μg/mL, 0.25 μg/mL, and 0.5 μg/mL were selected, and 100 μL of the prepared bacteria solution was added to the wells and incubated at a constant temperature of 35 °C for 16–20 h. To quantify the turbidity of the bacteria solution, we incubated the bacteria using five tigecycline drug-sensitive reagent plates, removed the bacteria solution from all the tigecycline drug-sensitive reagent plates, combined the bacteria solution of the same antibiotic concentration and diluted them at a ratio of 1:12, then put them into the bacterial turbidity meter for measurement.

## 3. Results and Discussion

### 3.1. Centrifugation

To characterize the effect of centrifugation, we took images of the bacteria in the bacteria solution in the counting plate before and after centrifugation. Before imaging, *E. coli* was first incubated overnight, and a bacteria solution of 0.5 MCF was adjusted using a bacterial turbidity meter and diluted in the drug sensitive inoculation medium at a concentration of 1:200. Two counting plates were taken, and 100 μL of the diluted bacteria solution was punched into each; one counting plate was not centrifuged whereas the other was. The bacteria in the bacteria solution were photographed at an interval of 100 μm upward along the depth direction, starting from the bacteria at the bottom of the counting plate. The images before and after centrifugation were processed, the area occupied by bacteria was counted, and a histogram was drawn (Figure 3). We can see that before centrifugation, the bacteria in the counting plate were distributed at different depths; after centrifugation, the vast majority of bacteria were enriched at the bottom of the counting plate, with only a small number of bacteria at other depths.

### 3.2. Images of Bacteria at Different Antibiotic Concentrations

The images of *E. coli* growth over time at different tigecycline concentrations were obtained according to the above scheme (Figure 4). As we can see from the images, with the increase of time, *E. coli* appeared to grow significantly without antibiotics and at 0.064 μg/mL concentration, specifically in terms of an increase in number and size; a small amount of bacteria grew in the bacteria solution at 0.125 μg/mL antibiotic concentration; while at 0.25 μg/mL and above, the growth of *E. coli* was significantly inhibited and basically did not grow again.

### 3.3. Image Processing Results

After acquiring the phase contrast image of *E. coli* growing under the effect of different concentrations of tigecycline over time, the *E. coli* was segmented by image processing. As shown in Figure 5, the *E. coli* phase contrast image was segmented by background subtraction, Gaussian filtering and adaptive thresholding, and it can be seen that the bacteria and the background are clearly separated, where the black is the bacteria and the white is the background. After image processing, the bacteria and background can be effectively segmented, which provides the conditions for a subsequent calculation of bacterial growth rate.

### 3.4. Change in Bacterial Growth Rate

Using the segmented image in 3.3, the *E. coli* area was obtained, the growth rate of *E. coli* was calculated according to Equation (1) and the growth rate change curve of *E. coli* was plotted (Figure 6). From the figure, it can be seen that the growth rate of *E. coli* gradually increased in the absence of antibiotics and low concentration of antibiotics, and gradually showed exponential growth. With the increase of antibiotic concentration, the growth rate increase of *E. coli* became smaller or even remained unchanged, indicating that the growth of *E. coli* was gradually inhibited with the increase of antibiotic concentration. The lowest antibiotic concentration that can significantly inhibit the growth of microorganisms is MIC [8], and it can be seen from the growth rate curve that bacterial growth is inhibited at 0.125, 0.25 and 0.5 μg/mL concentrations. Therefore, among the five concentrations we selected, the MIC is 0.125 μg/mL.

Therefore, we established a complete method on rapid AST of bacteria by the above means, i.e., centrifugation-single field of view tracking-image processing. Using this method, we can obtain the growth of bacteria over time in the presence of different concentrations of antibiotics and thus obtain the MIC of antibiotics.

### 3.5. Results of the Broth Micro-Dilution Method

The tigecycline drug-sensitive reagent plates were removed from the air bath thermostatic oscillator after incubation at 35 °C for 16–20 h, and the inhibition of *E. coli* was judged by the turbidity of the bacteria solution. After measuring the turbidity of the bacteria solution for all antibiotic concentrations, the turbidity of the bacteria solution at 0.5 μg/mL concentration was used as the reference, and the turbidity value was set to 0 to obtain the values of turbidity of the bacteria solution at different concentrations (Figure 7). From the figure, it can be seen that the turbidity of the bacteria solution was close to 0 at the antibiotic concentrations of 0.125 and 0.25 μg/mL, indicating that the growth of *E. coli* was inhibited at the antibiotic concentrations of 0.125 μg/mL and above, while the turbidity of the bacteria solution was larger under the effect of two antibiotic concentrations of 0 and 0.064 μg/mL, which can clearly demonstrate the massive growth of *E. coli* under the effect of these two concentrations of antibiotics. Based on the turbidity results, we can conclude that the 0.125 μg/mL concentration is the MIC of tigecycline acting on *E. coli*. Figure 7 also shows the results of our rapid AST method at hour 4, and it can be seen that the growth of *E. coli* under the action of different concentrations of tigecycline is consistent between the two methods, and the MIC results of the final determination are the same. We also used our method to image *E. coli* at different concentrations of tigecycline at hour 24 (Appendix A) and calculated the growth rate of *E. coli* (Appendix A), obtaining the same MIC of 0.125μg/mL.

### 3.6. Discussion

Our proposed method of tracking bacterial growth in a single field of view of centrifuged bacteria solution with the help of a phase contrast microscope and image processing enables us to obtain the MIC within 4 h. Also, we can achieve high throughput detection of bacteria–antibiotic combinations by increasing the number of wells in the counting plate mount. In this way, multiple MICs of bacteria–antibiotic combinations can be detected at the same time.

The background subtraction processing can effectively improve the uneven background of the phase contrast images by extracting the phase contrast image background and removing it from the original image. If adaptive thresholding is performed directly, some black areas in the background are easily segmented as bacteria; therefore, the accuracy of thresholding can be effectively improved by removing the background.

Bacteria grow too much over time and stack together, making it difficult to image all bacteria clearly, which has an impact on post-processing and counting bacterial growth rates, and therefore increases the standard deviation of experimental results to some extent.

## 4. Conclusions

In this study, we built our own centrifuge suitable for counting plates, which can achieve a rapid settling of bacteria to the bottom of the counting plate, so that bacteria can be enriched at the bottom surface and keep continuous growth at the focal plane of the phase contrast microscope, effectively reducing the effect of floating bacteria settling during growth and providing conditions for monitoring continuous growth of bacteria over time. In order to achieve a single field of view to track the growth of bacteria at multiple antibiotic concentrations, an automated phase contrast microscope was built independently. The proposed rapid AST method for liquid medium, i.e., centrifugation-single field tracking-image processing, was able to obtain MIC results within 4 h, and the experimental results were consistent with those of the broth micro-dilution method, validating the effectiveness of our method. Compared with other methods for rapid AST, our proposed phase contrast image-based method for rapid AST of liquid culture media is less costly and easy to implement, suitable for large-scale promotion, and also has the potential to achieve high-throughput rapid AST.

## Figures and Tables

**Figure 1 sensors-23-00059-f001:**
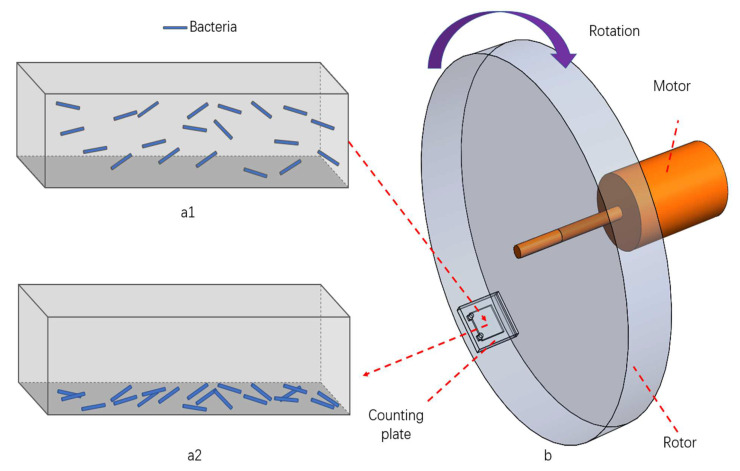
Schematic diagram of centrifugation effect. (**a1**) Before centrifugation: bacteria floating in the bacteria solution in the inner chamber of the counting plate; (**a2**) after centrifugation: bacteria settling on the bottom surface of the counting plate; (**b**) centrifuge: the motor drives the rotor to rotate and the counting plate is located at the edge of the rotor.

**Figure 2 sensors-23-00059-f002:**
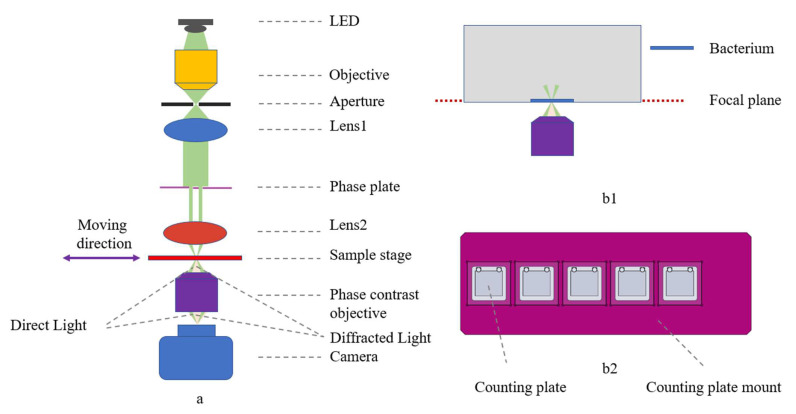
Phase contrast microscope overview. (**a**) Optical layout with ray tracing: direct light in green, diffracted light in orange; (**b1**) bacteria at the focal plane of the microscope after centrifugation; (**b2**) counting plate mount with five holes, allowing simultaneous detection of bacterial growth at five different concentrations of antibiotics.

**Figure 3 sensors-23-00059-f003:**
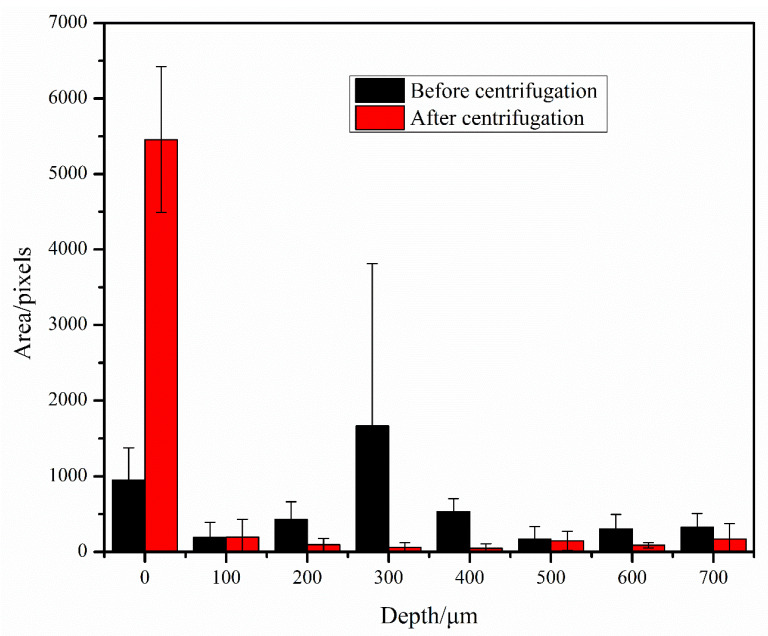
Bacterial area at different depth layers before and after centrifugation.

**Figure 4 sensors-23-00059-f004:**
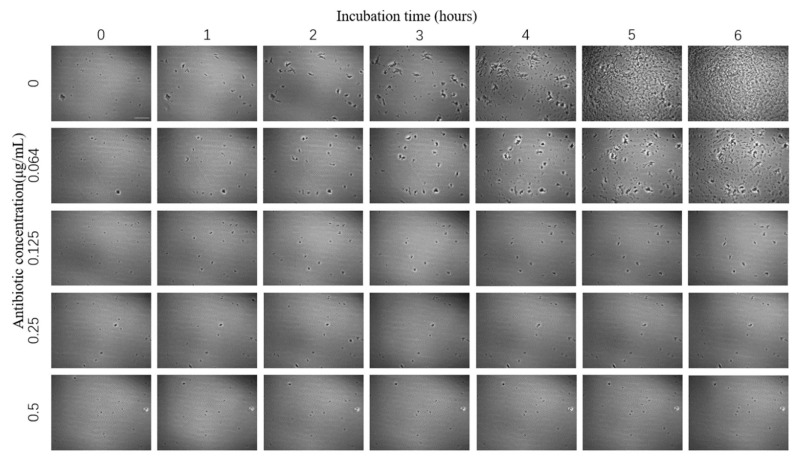
Images of *E. coli* at different times under the effect of different concentrations of tigecycline. Scale bar, 30 μm.

**Figure 5 sensors-23-00059-f005:**
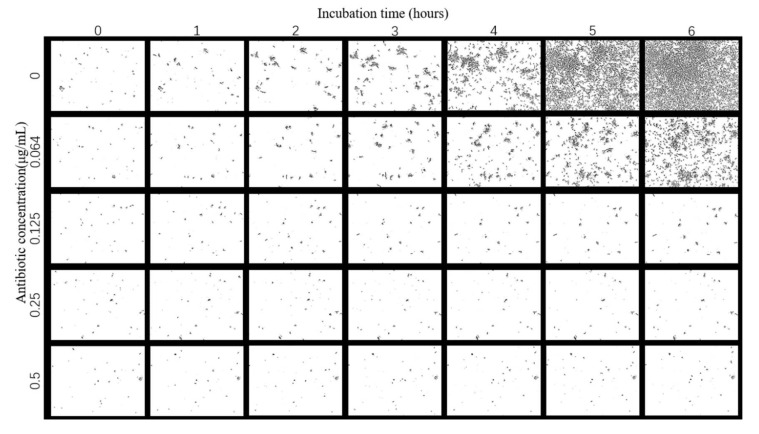
Segmentation results.

**Figure 6 sensors-23-00059-f006:**
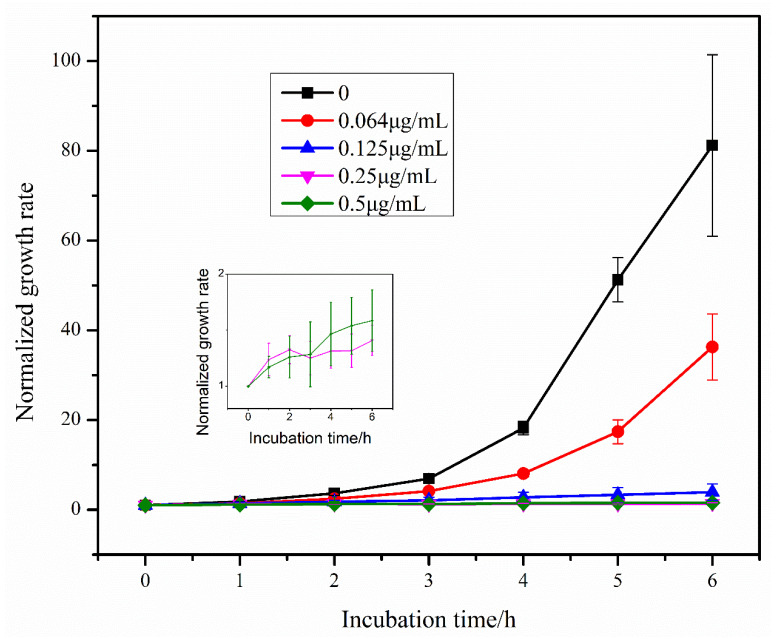
Growth rate of *E. coli* over time under the effect of different concentrations of tigecycline. Since the growth rate of *E. coli* did not change significantly under the effect of 0.25 μg/mL and 0.5 μg/mL tigecycline, we show the growth rate curves of *E. coli* under the effect of these two concentrations of tigecycline separately in the figure.

**Figure 7 sensors-23-00059-f007:**
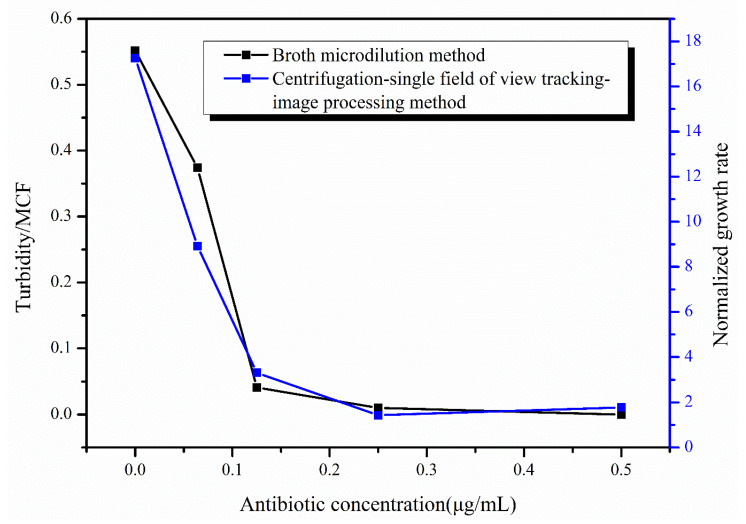
Results of the broth micro-dilution method (in black) and the centrifugation-single field of view tracking-image processing method (in blue).

## Data Availability

The data presented in this study are available on request from the corresponding author.

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
