# Peer review of "Phase Contrast Image-Based Rapid Antimicrobial Susceptibility Testing of Bacteria in Liquid Culture Media"

_sensors, 2022, doi:10.3390/s23010059_

Round 1

Reviewer 1 Report (Previous Reviewer 2)

The authors addressed all previous concerns and the paper is now significantly improved.

Author Response

Thank you for your review.

Reviewer 2 Report (Previous Reviewer 1)

The authors have revised it significantly by addressing the concerns of the reviewers and adding it as supplementary material. In my opinion now it can be accepted for publication. However, I would like that the following minor issues should be corrected/resolved.

Fig. 6, data points as well as growth rate curve for 0.25 microgram/mL are not distinguishable or missing.

L171, sentence can be rewritten as-'Normalized growth rate of bacteria is defined as[15].'

L196, The sentence can be modified  to "one counting plate was not centrifuged whereas the other was'.

Author Response

This manuscript is a resubmission of an earlier submission. The following is a list of the peer review reports and author responses from that submission.

Round 1

Reviewer 1 Report

In the present work, the authors report study MIC of bacteria under the effect of antibiotics using a hand made centrifuge and phase contrast microscopic system. In principle, the work can be published in 'Sensors'. However, I would like that they  should revise it considering the following suggestions.

In the conclusion they mention that "Compared with other methods for rapid AST, our proposed phase contrast image-based method for rapid AST of liquid culture media is less costly and difficult to implement, suitable for large-scale promotion, and also has the potential to achieve high-throughput rapid AST". I would suggest that they should provide a comparison (preferably in tabular form) of various methods (see reference 13) to validate their claim.

In Fig. 3, bacterial area increases even at 700 micro-meter depth. They should comment on this.

Fig. 7, Details of these two methods should be mentioned in the caption.

They have compared their results with broth micro-dilution method. They claim that their method is rapid. If other adopted methods (ref. 13) are not fast?

Give more details of 'upper computer. The word 'upper computer' word appears vague.

Why motor rotation is chosen as  '6400 rpm for 20s'? What happens in other cases?

Why Gaussian fitting is adopted for image processing?

In the conclusion section , 'difficult to implement' appears inappropriate.

There  are some language/wording related issues also e.g. line 41, use of article 'the'; line 73, in place of 'centrifuge to centrifuge',i tis better to use 'centrifuge to confine'; line 230 ' in place of 'appeared' it would have been' showed'; reference 9, i should in capital; reference 13, Journal name should be 'Sens. Actuators Reports'.

Reviewer 2 Report

All comments are given in the attached word document. Please refer to it.
